# Comparative Genomic Analysis of *Cutibacterium* spp. Isolates in Implant-Associated Infections

**DOI:** 10.3390/microorganisms11122971

**Published:** 2023-12-12

**Authors:** Anja Erbežnik, Andraž Celar Šturm, Katja Strašek Smrdel, Tina Triglav, Polona Maver Vodičar

**Affiliations:** Institute of Microbiology and Immunology, Faculty of Medicine, University of Ljubljana, 1000 Ljubljana, Slovenia; anja.erbeznik@mf.uni-lj.si (A.E.); andraz.celar-sturm@mf.uni-lj.si (A.C.Š.); katja.strasek@mf.uni-lj.si (K.S.S.); tina.triglav@mf.uni-lj.si (T.T.)

**Keywords:** *Cutibacterium* spp., *C. acnes*, *C. avidum*, *C. granulosum*, *C. namnetense*, implant-associated infections, virulence factors, virulence-associated genes, whole-genome sequencing

## Abstract

Bacteria of the genus *Cutibacterium* are Gram-positive commensals and opportunistic pathogens that represent a major challenge in the diagnosis and treatment of implant-associated infections (IAIs). This study provides insight into the distribution of different sequence types (STs) of *C. acnes*, and the presence of virulence factors (VFs) in 64 *Cutibacterium* spp. isolates from suspected or confirmed IAIs obtained during routine microbiological diagnostics. Fifty-three *C. acnes*, six *C. avidum*, four *C. granulosum*, and one *C. namnetense* isolate, collected from different anatomical sites, were included in our study. Using whole-genome sequencing and a single-locus sequencing typing scheme, we successfully characterized all *C. acnes* strains and revealed the substantial diversity of STs, with the discovery of six previously unidentified STs. Phylotype IA_1_, previously associated with both healthy skin microbiome and infections, was the most prevalent, with ST A1 being the most common. Some minor differences in STs’ distribution were observed in correlation with anatomical location and association with infection. A genomic analysis of 40 investigated VFs among 64 selected strains showed no significant differences between different STs, anatomical sites, or infection-related and infection undetermined/unlikely groups of strains. Most differences in VF distribution were found between strains of different *Cutibacterium* spp., subspecies, and phylotypes, with CAMP factors, biofilm-related VFs, lipases, and heat shock proteins identified in all analyzed *Cutibacterium* spp.

## 1. Introduction

Representatives of the genus *Cutibacterium* are Gram-positive, commensal, lipophilic, facultative anaerobic, and slow-growing rod-shaped bacteria [1]. As members of normal skin microbiota, they are often considered laboratory contaminants [2]. However, they are increasingly recognized as important opportunistic pathogens that can cause serious infections such as implant-associated infections (IAIs) [3,4,5,6], especially in immunocompromised individuals or when introduced into sterile areas during surgical procedures [7].

Bacteria of the genus *Cutibacterium* are intrinsically resistant to metronidazole but generally susceptible to clindamycin, macrolides, β-lactams, fluoroquinolones, and tetracycline. However, resistance to clindamycin, tetracycline, and macrolides is emerging [4,7]. Together with biofilm development on implant surfaces [8], the extended incubation period required to detect *Cutibacterium* spp., the frequent risk of contamination [9], and the absence of local or systemic signs of the infection [7] contribute to the challenging diagnosis and treatment of IAIs caused by *Cutibacterium* spp.

In addition to the best-known species, *Cutibacterium acnes*, there are several other *Cutibacterium* species (e.g., *C. avidum*, *C. granulosum*, *C. namnetense*, and *C. humerusii*), whose detection rates in clinical samples have increased significantly over the last decade due to improved microbiological diagnostics such as the sonication of removed medical devices, incubation period up to 14 days, and use of matrix-assisted laser desorption ionization-time of flight mass spectrometry (MALDI-TOF MS) identification [1,10,11,12].

Members of the genus *Cutibacterium* are generally considered to be opportunistic pathogens with low virulence, causing subacute or chronic infections. However, in 2008, Million et al. addressed the sepsis potential of *Cutibacterium* spp., particularly *C. avidum* [13], which differs from the other members of the genus in its specific tropism for a moist environment on the skin surface. On the contrary, *C. acnes* and *C. granulosum* require a lipid-rich skin surface [2,12,14,15,16] and are therefore less commonly associated with acne patients [16]. There is limited information on IAIs caused by *C. granulosum* and *C. namnetense*, with the latter being more similar to *C. acnes* than to other *Cutibacterium* species [4,6,14,17].

The members of *Cutibacterium* spp. (formerly known as *Propionibacterium* spp.) have been reclassified several times: in 1909, they were first recognized as members of the genus *Bacillus*, later as *Corynebacterium* spp., and only in 1946 as the genus *Propionibacterium*. The most recent reclassification occurred in 2016, when all cutaneous *Propionibacterium* spp. (*C. acnes*, *C. avidum*, *C. granulosum*, *C. namnetense*, and *C. humerusii*) were placed in a new genus called *Cutibacterium* spp. [4,14,16]. Currently, three subspecies and six phylotypes of *C. acnes* are known: *C. acnes* subsp. *acnes* (phylotypes IA_1_, IA_2_, IB, IC), *C. acnes* subsp. *defendens* (phylotype II), and *C. acnes* subsp. *elongatum* (phylotype III) [1,18,19]. Based on an analysis of the cell wall and the non-ribosomal housekeeping genes *recA* and *tly*, *C. acnes* strains were initially classified into two main types, I and II, followed by the additional type III, corresponding to strains with filamentous appendages [20,21,22]. To further increase the discriminatory power, multi-locus sequence typing (MLST) schemes have been developed, further classifying phylotypes into clonal complexes (CC) based on four (MLST_4_) [23], nine (MLST_9_—Aarhus scheme) [24], or eight housekeeping genes (MLST_8_—Belfast scheme) [25]. The development of a new SLST (single-locus sequencing type) scheme in 2014 divided *C. acnes* strains into clades IA-1, IA-2, IB-1, IB-2, IB-3, IC, II, III, which are further classified into sequence types (STs), of which 188 are currently known [26]. In 2023, a novel bi-locus sequence typing scheme CUTIS-SEQ was introduced, combining SLST and *camp2* loci [19].

The first complete genome sequence of *C. acnes* published in 2004 revealed a single circular chromosome with a size of 2,560,265 base pairs, corresponding to 2333 potential genes [27]. The differences in the genome size between different *Cutibacterium* spp. are not substantial, except for *C. granulosum*, which has a slightly smaller genome size (2.18 Mbp) than others.

Several previous studies have indicated a possible association between different phylotypes and specific clinical manifestations or anatomical sites. Phylotypes IB and II were commonly associated with IAI, soft tissue infections, and bacteremia, and were considered commensals of healthy skin [7,25,28]. Phylotype IA was mostly associated with acne, but the latest reports also report its presence in prosthetic joint infections [1]. Phylotype III is more commonly found on the trunk [1]. In spinal instrument infections, phylotype IA, especially CC18 (MLST_9_) and CC28 (MLST_9_) or ST A and F (SLST), were the predominant types [28,29]. Thus, these results are contradictory and may indicate that there is no predominant disease-specific phylotype [7]. No classification is currently available for other species of the genus *Cutibacterium*, likely due to the fact that the proportion of infections caused by *C. granulosum* (2.4–4.8%) and *C. avidum* (4.2–10.7%) among all *Cutibacterium* infections is low, and *C. avidum* is present in a minor proportion in skin microbiota (0.4%) compared to *C. acnes* [30,31,32]. Infections with *C. namnetense* and *C. humerusii* have rarely been reported. Several recent studies identified putative virulence factors (VFs) and virulence-associated genes important for environmental adaptation, adherence to target cells, enzymatic degradation of host tissues, and especially bacterial biofilm [1]. Biofilm production was observed in vivo and in vitro in several studies [7,8]. Kuehnast et al. suggested a possible correlation between the *C. acnes* phylotype and biofilm production rather than with the anatomical site of infection [33]. However, data on putative VFs in other *Cutibacterium* spp. are limited [1]. Therefore, the aim of this study was to perform an in silico comparative genomic analysis of 64 *Cutibacterium* spp. isolates from confirmed or suspected IAIs showing different clinical manifestations and isolated from different anatomical sites, with an emphasis on the presence of VFs and virulence-associated genes among the different *Cutibacterium* spp.

## 2. Materials and Methods

### 2.1. Sample Selection

A genomic analysis of *Cutibacterium* spp. strains isolated from patients with confirmed or suspected IAIs and surgical site infection (SSI) was performed at the Institute of Microbiology and Immunology, Faculty of Medicine, Ljubljana, Slovenia. We retrospectively reviewed our archive collection which included strains from 2012 to 2022, and in total, 64 *Cutibacterium* spp. strains associated with infections at different anatomical sites were selected (53 *C. acnes*, 6 *C. avidum*, 4 *C. granulosum* from IAIs, and 1 *C. namnetense* strain isolated from SSI).

Among 64 strains, 51 strains were obtained from the sonicate fluid (SF) of removed implants and 13 strains from peri-implant tissue (PT) samples from different anatomical locations: hip (19 samples; 16 SF, 3 PT), knee (14 samples; 11 SF, 3 PT), shoulder (4 samples; 4 SF), spine (13 samples; 11 SF, 2 PT), ankle (2 samples; 1 SF, 1 PT), 8 samples from the cardiovascular system (1 prosthetic valve; 2 cardiovascular implantable electronic devices—CIED, 1 ventricular assist devices—VAD; 1 stent graft; 1 femoral popliteal bypass; 2 coronary stents—CST), 1 breast implant, 2 central nervous system devices (CNSD), and one SSI strain.

The strains were defined as infection-related or infection-undetermined/unlikely. Infection-related strains were defined as such if they met the following microbiological criteria for IAIs: at least two positive PT cultures or positive SF culture and one PT culture with the same microorganism identified, or if more than ≥50 CFU/mL was detected in the SF by conventional culture methods [34]. When detailed clinical data were not available, strains were characterized as infection undetermined/unlikely, since the criteria mentioned above may miss cases with clinical signs suggestive of infection but inconsistent with the microbiological results. The study protocol was approved by the Slovenian National Medical Ethics Committee of Slovenia (Ministry of Health, Republic of Slovenia) under identification number 0120-15/2022/6 (date of approval: 31 May 2022).

### 2.2. DNA Isolation

Isolates were collected from frozen stocks stored at −80 °C and incubated anaerobically on Schaedler agar plates at 35 °C for 72 h. Identification was confirmed using MALDI-TOF MS. Total DNA was extracted using the DNeasy Blood & Tissue Kit (Qiagen Ltd., West Sussex, United Kingdom) according to the manufacturer’s protocol for Gram-positive bacteria. The extracted DNA was stored at −80 °C until further use. A Qubit 3.0 fluorometer in combination with a Qubit 1× dsDNA HS assay kit (Thermo Fisher Scientific, Waltham, MA, USA) were used to determine the amount of DNA. In addition, DNA purity was assessed based on the absorbance ratio at A_260/280_ and A_260/230_ ratios using a NanoDrop 2000/2000c spectrophotometer (Thermo Scientific, Waltham, MA, USA). Only DNA samples with a concentration higher than 1 ng/µL and an A_260/280_ ratio between 1.8 and 2.0 were selected for whole-genome sequencing (WGS) and included in further analysis.

### 2.3. Whole-Genome Sequencing

WGS of 64 *Cutibacterium* spp. strains was performed at the Institute of Microbiology and Immunology, Faculty of Medicine, University of Ljubljana with Illumina NextSeq 550 (Illumina, San Diego, CA, USA) using a paired-end (2 × 150 bp) sequencing protocol. Strains were sequenced to a minimum coverage of 150×. A Nextera XT DNA Library Preparation Kit (Illumina, San Diego, CA, USA) was used for DNA library preparation. Raw reads were trimmed for adapter sequences and low-quality reads with Fastp v0.23.2 [35]. The quality of raw and trimmed reads was assessed with FastQC v0.11.9 [35]. Trimmed reads were assembled into contigs using SPAdes v3.15.3 [36] and default *k*-mer values and “--careful” parameters. Quast v5.2.0 [37] was used for a quality assessment of the assemblies. The minimum threshold for assembly quality was set at a value of 20,000 bp for *N_50_* and the total number of contigs lower than 500. Prokka v1.14.6 [38] was used for annotation of the bacterial genomes, using the genome assembly of *C. acnes* strain HL096PA1 [39] (*C. acnes*) as the reference genome. The assembled genomes were further analyzed using Ridom SeqSphere v9.0.10 [40] and zDB software v1.1.1 [41].

### 2.4. SLST and Phylogenetic Analysis

For the *C. acnes* strains, the STs were determined in silico using the SLST database from Aarhus University (http://www.medbac.dk/slst_server_script.html, accessed on 27 July 2023).

A pan-genome analysis based on the accessory genome of 64 *Cutibacterium* spp. strains was performed using Roary v3.13.0 [42] and used for further phylogenetic analysis. A phylogenetic tree was constructed using FastTree v2.1.10 according to the varieties in groups of genes uniquely present/absent in accessory genome and visualized using iTOL v6.3 (https://itol.embl.de/, accessed on 24 July 2023) [43].

### 2.5. Analysis of Virulence-Associated Genes and Antimicrobial Resistance Genes

ABRicate v1.0.1 (https://github.com/tseemann/abricate, accessed on 10 May 2023) was used for the analysis of virulence factors by creating a custom database (Appendix A) consisting of 40 VFs collected from the literature (Table 1). Additional species–specific sequences were analyzed in silico using blastn_ffa, tblastn, and blastp functions within the zDB tool. KEGG (Kyoto Encyclopedia of Genes and Genomes) Ortholog annotations, COG (Clusters of Orthologous Genes) annotations, or Pfam domains of VFs and their homologues were identified with the zDB software. The antimicrobial resistance genes present were identified with the AMRFinderPlus v3.11.2 database [44].

## 3. Results

### 3.1. Sequence Typing and Association with Clinical Relevance

The sequence typing results are summarized in Table 2 and in Figure 1 and Figure 2. All *C. acnes* strains were successfully assigned to specific STs using the SLST scheme. In addition to the already known STs, six new STs were identified, namely A60, A61, C9, H18, K36, and K37. In our cohort, the most common phylotype among *C. acnes* strains was IA_1_, which was identified in 62% (33/53) of cases, followed by phylotype II in 19% (10/53), and phylotype IB in 15% (8/53). Phylotypes IA_2_ and III were both represented by a single strain.

We observed no significant difference in the distribution of phylotypes among the infection-related and infection-undetermined/unlikely strains (Table 2). Interestingly, *C. acnes* ST A2, A61, C1, C9, and K2 were found exclusively in the infection-undetermined/unlikely group of strains.

### 3.2. Phylogenetic Analysis

The WGS results are summarized in Appendix A and run accession numbers for each strain in Appendix A. The initial genome screening of the 53 *C. acnes* strains revealed no significant overall differences in genome size (2.49 ± 0.09 Mbp; mean ± SD) or GC content (60%). Phylotype III (2.57 Mbp) had a slightly larger genome than the phylotypes IB (2.55 Mbp ± 0.02) and IA_1_ (2.52 Mbp ± 0.03). Among the other *Cutibacterium* spp., *C. avidum* (2.5 Mbp ± 0.02) and *C. namnetense* (2.41 Mbp) had a similar genome size, whereas *C. granulosum* (2.17 Mbp ± 0.05) had the smallest genome size. *C. granulosum* had the highest GC content with 64%, followed by *C. avidum* with 63% and *C. namnetense* with 61%. The number of putative protein coding sequences in the genomes varied from 1762 (*C. granulosum*) to 2464 (*C. acnes* subsp. *elongatum*), with an average of 2335.36 ± 33.81 in *C. acnes*, 2266.17 ± 26.49 in *C. avidum*, 2218 in *C. namnetense*, and 1805.5 ± 30.12 in *C. granulosum*. To assess the genomic relatedness of strains belonging to different *Cutibacterium* species, the presence and absence of genes according to a pan-genome analysis were determined. A total of 11,770 coding sequences were identified.

### 3.3. Antimicrobial Resistance Genes

*Cutibacterium* spp. isolates are intrinsically resistant to metronidazole. Phenotypically, we detected resistance to clindamycin using the gradient diffusion method in two isolates (CUTI-242-14 and CUTI-260-33) during routine microbiological diagnostics. Both strains contained the *erm*(X) gene associated with clindamycin resistance. In addition, we found *erm*(X) in one strain (CUTI-260-32) which was phenotypically susceptible to clindamycin. No other antimicrobial resistance genes were identified, which was in perfect agreement with the phenotypical testing.

### 3.4. Analysis of Putative Virulence Factors

We investigated the presence of 40 putative VFs in 64 *Cutibacterium* spp. strains with the aim of determining their distribution in different *Cutibacterium* species, subspecies, phylotypes, STs, anatomical sites, and association with infection. Most of the selected VFs (Table 1 and Appendix A) and their associated genes that were analyzed in this study were previously identified and described in *C. acnes* strains. This could potentially lead to the misidentification of homologs that were identified based on whether they belonged to the same KEGG ortholog annotation, COG annotation, or Pfam domain (Appendix A). We characterized homologs into four subgroups according to the percentage of identity: homologs with very high identity (>95%), homologs with high identity (>80% and <95%), homologs with low identity (>60% and <80%), and homologs with very low identity (<60%) (Appendix A). The VFs’ distribution in different *Cutibacterium* spp. strains is summarized in Figure 1. *C. acnes* and *C. namnetense* had the highest number of identified VFs and most similar distribution of VFs, whereas *C. granulosum* had the lowest number of identified putative VFs and the most diverse distribution compared with other *Cutibacterium* spp. The present results did not show the specific distribution of VFs in association with particular anatomic sites or clinical manifestations. The main differences in the presence of VFs were observed primarily at the level of the *Cutibacterium* species, subspecies, and phylotypes.

All relevant references and previous reported distribution of VFs are listed in Table 1 and Appendix A.

## 4. Discussion

We successfully characterized all selected *C. acnes* strains and identified six new STs in our study cohort: A60, A61, C9, H18, K36, and K37. Consistent with previous studies, phylotype IA_1_ was found to be the most prevalent, with ST A1 being the most common [1,49,50].

Phylotype IA_1_ has been considered a common commensal in a healthy skin microbiota and associated with severe acne [24,29,51,52,53], but is becoming increasingly recognized as the predominant phylotype in IAI infections [7,23,25,29,54]. In the past, phylotype IB was the one most commonly associated with IAI. Among the *C. acnes* strains, the group of strains characterized as phylotype IB, had with 63% (5/8) the highest percentage of infection-related strains in our cohort. Certain STs were found exclusively in the infection-indeterminate/unlikely group of strains. Although some minor differences in specific STs were seen in association with the anatomic location and association with infection among the different STs, any correlation between STs, anatomic location, or association with infection was inconclusive due to limited clinical data and sample sizes.

No significant differences in the presence of VFs and their associated genes were observed between the different STs of *C. acnes*, different anatomic sites, or between the infection-related and infection indeterminate/unlikely group of strains. However, differences in the presence of VFs were detected at the level of bacterial species and, in the case of *C. acnes*, also at the level of subspecies and phylotypes. The most prominent differences in the gene presence were observed for hyaluronate lyase (*HYL*) genes, which encode enzymes responsible for the degradation of hyaluronic acid in the host extracellular matrix, thereby facilitating bacterial invasion and tissue colonization, particularly *HYL-IB/II* and *HYL-IA*. *HYL-IB/II* was only detected in phylotypes IB and II, whereas *HYL-IA* was found exclusively in phylotype IA. This result is in contrast to previous reports showing the presence of *HYL-IA* in phylotypes IA and IB and *HYL-IB/II* in phylotypes IA, IB, and II [1]. Another interesting observation is that *HYL-IA* was detected in one *C. namnetense* strain and *HYL-IB/II* in one *C. granulosum* strain (CUTI-243-32) isolated from a hip prosthesis infection.

The *clpS* gene, which encodes the Clp protease adaptor protein essential for the control of intracellular protein degradation, has been observed in a previous study mainly in IA phylotypes [46], which is in agreement with the present study where it was identified in phylotypes IA, phylotype III, and in *C. avidum* strains. Remarkably, we did not find it in phylotypes II, IB, *C. granulosum*, and *C. namnetense*. The *clpS* gene was also detected in the non-coding region in other *C. acnes* strains, characterized by the insertion of nucleotide T at position 129. We hypothesize that this insertion could potentially cause a frameshift mutation that disrupts the reading frame and results in the absence of the protein product.

Heat shock proteins (HSPs) play a critical role in various prokaryotes, performing functions in stress response, protein folding, intracellular survival, potential evasion of host immune responses, and more. All investigated HSPs and genes related to HSPs’ production (e.g., *dnaK*, *groEL*, *dnaJ*, *grpE*, GroES, and *hsp20*) were detected in all 64 strains [55,56]. In cases where the HSPs were not initially recognized as homologs, a further analysis confirmed the presence of species-specific HSPs (accession numbers in Appendix A).

All five CAMP factors previously known as Christie-Atkins-Munch-Petersen factors, responsible for triggering tissue damage by membrane pore formation, were found in all *C. acnes* strains analyzed, which is consistent with the previous findings [7,23,57]. *C. acnes*-specific CAMP factors 1, 3, 4, and 5 were additionally confirmed in the *C. namnetense* strain. CAMP factors 3 and 5 were identified in all species, but as homologs with a very low identity in *C. granulosum*. Other CAMP factors were absent in both *C. granulosum* and *C. avidum*. Instead, two previously described species-specific CAMP factors were identified in *C. granulosum* and *C. avidum* (accession numbers in Appendix A). Both CAMP genes in *C. avidum* showed a fairly high identity with the CAMP 3 and CAMP 5 genes of *C. acnes*, consistent with the conclusions of Mak et al. [45].

Numerous putative VFs may be associated with the process of biofilm formation, including cell envelope-related transcriptional attenuator, *rcsB*, *ytpA*, *flp*, *luxS*, YhjD/YihY/BrkB family envelope integrity protein, cell fate regulator YaaT, and, indirectly, probably the proteins GroES and CAMP 1. Biofilm-regulating protein A (BrpA), *C. granulosum*-specific VF, previously described in *Streptococcus* species, was detected in all our *C. granulosum* strains identified as the putative cell wall biosynthetic protein LcpB, as well as homologs with lower identity in other species (Appendix A). In contrast, homologs of the cell fate regulator YaaT, which controls sporulation, competence, and biofilm development, were found with high sequence identity in all *Cutibacterium* species from this study cohort, except for homologs in *C. granulosum*, which had a lower identity (Appendix A). The YhjD/YihY/BrkB family envelope integrity protein, originally found in *Bordetella pertussis*, was identified as VF BrkB in all *Cutibacterium* spp. strains, but the homologs in *C. granulosum* showed very low identity (Appendix A). A hypothetical protein common to all strains of phylotype IA was identified as a response regulator (*rcsB*), confirming the results of Cavallo et al. [46]. Putative adhesion proteins may also play a critical role in biofilm formation, as the previously described putative adhesin in the reference genome of *C. acnes* (HL096PA1) was identified in all strains, which may also play a critical role in biofilm formation. Putative adhesive protein homologs were not identified in *C. granulosum*, while in *C. avidum* and *C. namnetense* they were present as homologs with high identity. Genes for dermatan sulfate adhesion proteins *dsA1* and *dsA2* were absent in *C. avidum* and *C. granulosum*, in our study as well as in Mak et al., but we confirmed the presence of a low identity *dsA1* homolog in the strain of *C. namnetense* [45].

We identified the presence of the genes-encoding enzymes phospholipase *ytpA* and *luxS* and adhesive *flp* pili (TadE/G) in all *Cutibacterium* strains. Previous reports indicated the exclusive presence of *luxS*, which is involved in quorum sensing, in phylotypes IA_1_, IB, and II [46]; adhesive *flp* pili (TadE/G) in phylotype II [1]; and *ytpA* in phylotype IA_1_ [46].

The genus *Cutibacterium* has lipolytic properties; it possesses several lipases that can hydrolyze triglycerides into fatty acids. They were one of the first putative VFs identified because lipid degradation can promote inflammation. One of them, triacylglycerol lipase, occurs in two variants as *gehA* and *gehB*, with a 42% identity at the protein level between them [1], and the products, free fatty acids, are thought to contribute to the pathogenesis of acne. In this study, we confirmed the presence of *gehA* in all phylotypes as well as in *C. avidum* and *C. namnetense* as homologs with very high identity and *gehB* in all phylotypes and also as homologs with high identity in *C. avidum* and *C. namnetense.* We also identified three triacylglycerol lipases as potential homologs in *C. granulosum* (Appendix A).

Polyunsaturated fatty acid (PUFA) isomerase was absent in the strain of *C. acnes* subsp. *elongatum* and other *Cutibacterium* species [1]. The presence of this enzyme remains relatively poorly understood, and its role in *Cutibacterium* spp. virulence has not been extensively studied. This enzyme is involved in the production of short-chain fatty acids (SCFAs) produced by *C. acnes* during fermentative growth. These SCFAs include propionate, acetate, butyrate, and valerate, and they may be associated with the suppression of *S. aureus* growth [58], inhibition of *S. epidermidis* biofilm formation [59], and possible adverse effects on skin barrier functions [1,60].

In addition to lipid hydrolysis, *C. acnes* has several enzymes that can process glycolipids. Sialidase A and B (*nanA*, *nanB*) were present in all *C. acnes* strains and in *C. namnetense*, while absent in other *Cutibacterium* spp., except in one *C. avidum* strain (CUTI-216-55), characterized as infection undetermined/unlikely, where sialidase B was present as a homolog with a very low identity. Glycosidase was present only in two strains *C. granulosum* (CUTI-243-32 and CUTI-515-74). In the other species, it was present in all other *Cutibacterium* species, in two *C. avidum* strains as a homolog with a very low identity (CUTI-233-15 and CUTI-216-55), and in others as a homolog with a high identity. Lipashydrolase (*menH*) and endo-β-N-acetylglucosaminidase were identified in both *C. acnes* and *C. namnetense*. In addition, endoglycoceramidase, which has been previously described as a potential VF because of its presence in the infundibulum of sebaceous follicles [1,55], was found in *C. acnes*, *C. namnetense*, and *C. avidum*.

Previously, acetyl-CoA synthetase (*acsA*), which is important for lipid transport and metabolism, had been identified mainly in the IA_1_ phylotype. However, our study has now confirmed that very high identity homologs exist in all phylotypes of *C. acnes* and in *C. namnetense*. There are limited data available on several other VFs that were analyzed in this study. The shikimate kinase, encoded by *gntK*, has been identified in phylotypes IB, II, and III, and was previously reported only in the phylotypes IB and II [46]. In other *Cutibacterium* species, it was found as a homolog. RoxP, short for “resistance to oxidative stress protein P”, known for its role in protecting the bacterium from oxidative stress by reactive oxygen species produced by the host immune system, was detected only in strains of *C. acnes* and *C. namnetnese*. The surface protein transpeptidase—sortase F (*srtF*) was previously identified in all phylotypes, which was confirmed in our study. Its homologs as hypothetical proteins were found in *C. avidum* and in *C. granulosum* (Appendix A). Iron acquisition protein (*htaA*), a polyunsaturated fatty acid isomerase, was present in all *C. acnes* phylotypes and in *C. namnetense*. Additionally, it was identified as a species-specific homolog in *C. avidum*. The dipeptide transport system permease protein *dppB*, previously described in the IA_1_ phylotype [46], has been identified as *dppB_1* in *C. acnes*, *C. namnetense*, and *C. avidum* and as homolog *dppB_2* in *C. granulosum* and is important for amino acid transport and inorganic ion transport in metabolism. The repressor gene of porphyrin synthesis, *deoR*, was confirmed in all *C. acnes* phylotypes and other *Cutibacterium* spp. with a high identity.

## 5. Conclusions

While *C. acnes* is increasingly recognized as an opportunistic pathogen in IAIs, causing mostly low-grade and chronic infections, the role of other *Cutibacterium* species remains poorly understood. This study provides additional information on the genomic diversity of *C. acnes* strains and shows the distribution of VFs among 64 *Cutibacterium* spp. strains from IAIs. While some of the present results differ from the previously reported distributions of VFs in *C. acnes*, previous studies have mainly focused on strains from patients with acne, in contrast to our study, which focuses on implant-associated infections. While our study found some diversity in the prevalence of virulence factors (VFs) within the genus *Cutibacterium*, no significant differences in the presence of VFs were observed between different STs, different anatomical sites, or association with infection. Overall, the present results improve the understanding of the genetic diversity and virulence potential of *C. acnes* and other *Cutibacterium* species and emphasize the need for further studies to fully understand the specific relationships between different virulence factors, bacterial genotypes, and disease pathogenesis.

## Figures and Tables

**Figure 1 microorganisms-11-02971-f001:**
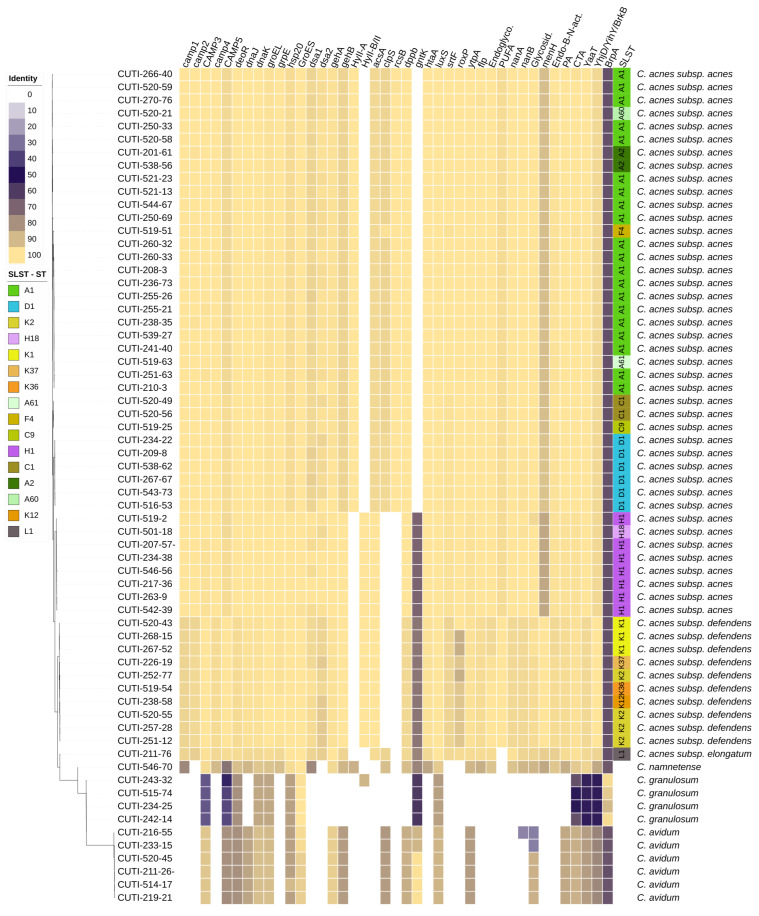
Phylogenetic tree of the accessory genome of 64 strains of *Cutibacterium* spp. The distribution of VFs is represented as a heat map showing the percentage of homology of putative, mostly *C. acnes*-specific, VFs. The tree is based on groups of genes uniquely present/absent in the accessory genome.

**Figure 2 microorganisms-11-02971-f002:**
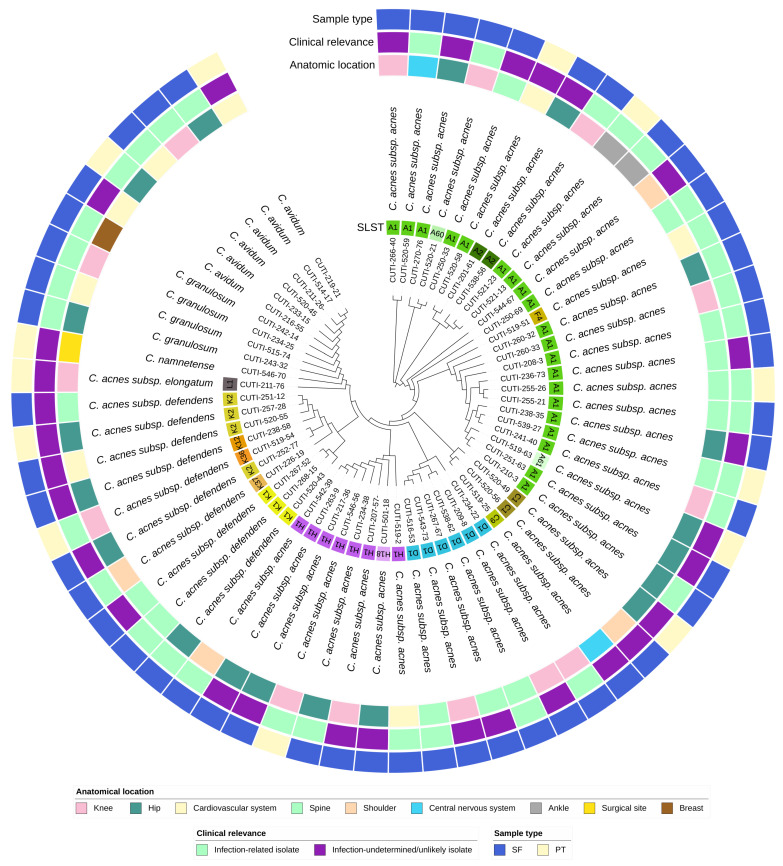
Circular phylogenetic tree of the accessory genome of 64 isolates of *Cutibacterium* spp. with ignored branch lengths. The tree is based on groups of genes uniquely present/absent in the accessory genome. The tree is annotated with the following strain metadata: SLST ST (for *C. acnes* strains), classification of strains as infection-related or infection-undetermined/unlikely, sample type (SF—sonication fluid; PT—peri-implant tissue), and anatomical location.

**Table 1 microorganisms-11-02971-t001:** The list of putative VFs and their distribution among phylotypes of *C. acnes* and other *Cutibacterium* spp. Where VF-associated genes were analyzed, VFs are written in italics.

	Reported Distribution among Phylotypes According to the Literature
CAMP factors	Christie-Atkins-Munch-Petersen factors
CAMP factor 1	Reported in all phylotypes of *C. acnes* [1,29]. Missing in *C. avidum* [45]. No data for *C. granulosum* and *C. namnetense*.
CAMP factor 2	Reported in all phylotypes of *C. acnes* [1,29]. Missing in *C. avidum* [45]. No data for *C. granulosum* and *C. namnetense*.
CAMP factor 3	Reported in all phylotypes of *C. acnes* [29]. According to Mayslich et al. only in phylotype II and III [1]. High % identity homologue of *C. avidum* CAMP factor [45]. No data for *C. granulosum* and *C. namnetense*.
CAMP factor 4	Reported in all phylotypes of *C. acnes* [29]. According to Mayslich et al. only in phylotype II and III [1]. No data for *C. granulosum* and *C. namnetense*.
CAMP factor 5	Reported in all phylotypes of *C. acnes* [29]. High % identity homologue in *C. avidum* [45]. No data for *C. granulosum* and *C. namnetense*.
Heat shock proteins	
*dnaJ*	Chaperone DnaJ, Hsp40 is reported in phylotypes IA_1_, IB, and II of *C. acnes* [46]. No data for *C. avidum*, *C. granulosum*, and *C. namnetense*.
*dnaK*	Chaperone DnaK, Hsp70 is reported in all phylotypes of *C. acnes* [1]. No data for *C. avidum*, *C. granulosum*, and *C. namnetense*.
*groEL*	Chaperone GroEL, Hsp60 is reported in phylotypes IA_1_, IB, and II of *C. acnes* [46]. No data for *C. avidum*, *C. granulosum*, and *C. namnetense*.
*grpE*	No data for bacterial nucleotide exchange factor in any *Cutibacterium* spp. species.
*Hsp20*	Chaperone Hsp20 is reported in all phylotypes of *C. acnes* [1]. No data for *C. avidum*, *C. granulosum*, and *C. namnetense.*
GroES	Chaperone GroES, Hsp10 is reported in all phylotypes of *C. acnes* [1]. No data for *C. avidum*, *C. granulosum*, and *C. namnetense.*
Biofilm formation	
*dsA1*	Dermatan sulphate adhesion 1 reported in phylotypes IA, IB, and II of *C. acnes* [1]. Absent in *C. avidum* and *C. granulosum* [45]. No data for *C. namnetense*.
*dsA2*	Dermatan sulphate adhesion 2 reported in phylotypes IA and IB of *C. acnes* [1]. Absent in *C. avidum* and *C. granulosum* [45]. No data for *C. namnetense*.
*rcsB*	Biofilm formation response regulator reported in phylotype IA_1_ of *C. acnes* [46]. No data for *C. avidum*, *C. granulosum*, and *C. namnetense*
*luxS*	VF involved in the quorum sensing reported in phylotypes IA_1_, IB, and II of *C. acnes* [46]. No data for *C. avidum*, *C. granulosum*, and *C. namnetense*.
YaaT	Cell fate regulator controlling sporulation, competence, and biofilm development. No data for any *Cutibacterium* spp. species.
*ytpA*	Phospholipase reported in phylotype IA_1_ of *C. acnes* [46]. No data for *C. avidum*, *C. granulosum*, and *C. namnetense.*
*Flp*	Pilus-assembly TadE/G reported in phylotype II of *C. acnes* [1]. No data for *C. avidum*, *C. granulosum*, and *C. namnetense.*
Putative adhesion protein	Putative adhesion protein reported in all phylotypes of *C. acnes* [27]. No data for *C. avidum*, *C. granulosum*, and *C. namnetense*.
YhjD/YihY/BrkB	YhjD/YihY/BrkB family envelope integrity protein. No data for any *Cutibacterium* spp. species.
*acsA*	Acetyl-CoA synthetase reported in phylotype IA_1_ of *C. acnes* [46]. No data for *C. avidum*, *C. granulosum*, and *C. namnetense*.
Lipases	
*GehA*	Triacylglycerol lipase A reported in all phylotypes of *C. acnes* [1]. No data for *C. avidum*, *C. granulosum*, and *C. namnetense*.
*GehB*	Triacylglycerol lipase B reported in all phylotypes of *C. acnes* [1]. No data for *C. avidum*, *C. granulosum*, and *C*. *namnetense*.
Other VFs	
*HYL-IA*	Hyaluronate lyase type A reported in phylotype IA and IB of *C. acnes* [1]. No data for *C. avidum*, *C. granulosum*, and *C. namnetense.*
*HYL-IB/II*	Hyaluronate lyase type B reported in phylotype IA, IB, and II of *C. acnes* [1]. No data for *C. avidum*, *C. granulosum*, and *C. namnetense*.
*clpS*	Clp protease adaptor protein reported in phylotype IA_1_ of *C. acnes* [46]. No data for *C. avidum*, *C. granulosum*, and *C. namnetense*.
*deoR*	Repressor gene of porphyrin synthesis reported in all phylotypes *of C. acnes* [1] and according to Cobain et al. only in phylotypes II and III [47]. According to Barnard et al. reported in *C. avidum* and *C. granulosum* [48]. No data for *C. namnetense*.
*Dppb*	Dipeptide ABC transporter reported in phylotype IA_1_ of *C. acnes* [46]. No data for *C. avidum*, *C. granulosum*, and *C. namnetense.*
*gntK*	Shikimate kinase reported in phylotypes IB and II of *C. acnes* [46]. No data for *C. avidum*, *C. granulosum*, and *C. namnetense*.
*htaA*	Iron acquisition protein reported in phylotypes IA_1_, IB, and II of *C. acnes* [46]. No data for *C. avidum*, *C. granulosum*, and *C. namnetense.*
*SrtF*	Sortase F—surface protein transpeptidase reported in all phylotypes of *C. acnes* [1]. No data for *C. avidum*, *C. granulosum*, and *C. namnetense*.
*RoxP*	Radical oxygenase is reported in all phylotypes of *C. acnes* [1]. No data for *C. avidum*, *C*. *granulosum*, and *C. namnetense.*
Endoglycoceramidase	Endoglycoceramidase is reported in all phylotypes of *C. acnes* [1,27]. No data for *C. avidum*, *C. granulosum*, and *C. namnetense.*
PUFA isomerase	Polyunsaturated fatty acid isomerase reported to be potentially present in all phylotypes of *C. acnes* [1]. No data for *C. avidum*, *C. granulosum*, and *C. namnetense.*
Sialidase *nanA and nanB*	Sialidase A and B are reported in all phylotypes of *C. acnes* [1]. No data for *C. avidum*, *C*. *granulosum*, and *C. namnetense*.
Glycosidase	Glycosidase is reported in all phylotypes of *C. acnes* [1]. No data for *C. avidum*, *C*. *granulosum*, and *C. namnetense*.
*menH*	Lipase hydrolase reported in phylotypes IA and IB of *C. acnes* [1]. No data for *C. avidum*, *C. granulosum*, and *C. namnetense.*
Endo-β-N-acetylglucosaminidase	Endo-β-N-acetylglucosaminidase reported in *C. acnes* [27]. No data for *C. avidum*, *C. granulosum*, and *C. namnetense*.
Cell envelope-related transcriptional attenuator	No data for any *Cutibacterium* spp. species.

**Table 2 microorganisms-11-02971-t002:** WGS results, sequence typing, and clinical relevance data.

Sample ID	Species and Subspecies	Phylotypes *	ST **	Source	Anatomic Location of Implant or Peri-Implant Tissue	Infection-Related or Infection Undetermined/Unlikely Strain
CUTI-243-32	*C. granulosum*	/	/	SF	Hip	Infection-related
CUTI-515-74	*C. granulosum*	/	/	SF	Prosthetic valve	Infection-related
CUTI-234-25	*C. granulosum*	/	/	SF	Knee	Infection-related
CUTI-242-14	*C. granulosum*	/	/	SF	Breast	Infection-related
CUTI-233-15	*C. avidum*	/	/	PT	Hip	Infection-related
CUTI-216-55	*C. avidum*	/	/	SF	Femoral popliteal bypass	Infection undetermined/unlikely
CUTI-211-26	*C. avidum*	/	/	SF	Knee	Infection-related
CUTI-520-45	*C. avidum*	/	/	SF	Stent graft	Infection-related
CUTI-219-21	*C. avidum*	/	/	PT	CST	Infection undetermined/unlikely
CUTI-514-17	*C. avidum*	/	/	SF	Hip	Infection-related
CUTI-546-70	*C. namnetense*	/	/	PT	Surgical site	Infection-related
CUTI-211-76	*C. acnes* subsp. *elongatum*	III	L1	PT	Knee	Infection undetermined/unlikely
CUTI-520-43	*C. acnes* subsp. *defendens*	II	K1	SF	Hip	Infection-related
CUTI-268-15	*C. acnes* subsp. *defendens*	II	K1	SF	Spine	Infection-related
CUTI-267-52	*C. acnes* subsp. *defendens*	II	K1	SF	Spine	Infection undetermined/unlikely
CUTI-252-77	*C. acnes* subsp. *defendens*	II	K2	SF	Hip	Infection undetermined/unlikely
CUTI-226-19	*C. acnes* subsp. *defendens*	II	K37	SF	Shoulder	Infection-related
CUTI-519-54	*C. acnes* subsp. *defendens*	II	K36	PT	Knee	Infection-related
CUTI-251-12	*C. acnes* subsp. *defendens*	II	K2	SF	Spine	Infection undetermined/unlikely
CUTI-238-58	*C. acnes* subsp. *defendens*	II	K12	SF	Hip	Infection-related
CUTI-520-55	*C. acnes* subsp. *defendens*	II	K2	PT	VAD	Infection undetermined/unlikely
CUTI-257-28	*C. acnes* subsp. *defendens*	II	K2	SF	Hip	Infection undetermined/unlikely
CUTI-519-2	*C. acnes* subsp. *acnes*	IB	H1	SF	CIED	Infection-related
CUTI-501-18	*C. acnes* subsp. *acnes*	IB	H18	SF	Hip	Infection undetermined/unlikely
CUTI-207-57	*C. acnes* subsp. *acnes*	IB	H1	SF	Knee	Infection undetermined/unlikely
CUTI-546-56	*C. acnes* subsp. *acnes*	IB	H1	PT	Knee	Infection-related
CUTI-234-38	*C. acnes* subsp. *acnes*	IB	H1	SF	Hip	Infection-related
CUTI-542-39	*C. acnes* subsp. *acnes*	IB	H1	SF	Shoulder	Infection-related
CUTI-217-36	*C. acnes* subsp. *acnes*	IB	H1	SF	Hip	Infection-related
CUTI-263-9	*C. acnes* subsp. *acnes*	IB	H1	SF	Hip	Infection undetermined/unlikely
CUTI-234-22	*C. acnes* subsp. *acnes*	IA_1_	D1	SF	Knee	Infection-related
CUTI-516-53	*C. acnes* subsp. *acnes*	IA_1_	D1	SF	Spine	Infection-related
CUTI-543-73	*C. acnes* subsp. *acnes*	IA_1_	D1	SF	Knee	Infection undetermined/unlikely
CUTI-538-62	*C. acnes* subsp. *acnes*	IA_1_	D1	SF	Spine	Infection-related
CUTI-267-67	*C. acnes* subsp. *acnes*	IA_1_	D1	SF	Spine	Infection undetermined/unlikely
CUTI-209-8	*C. acnes* subsp. *acnes*	IA_1_	D1	SF	Knee	Infection undetermined/unlikely
CUTI-519-51	*C. acnes* subsp. *acnes*	IA_2_	F1	SF	CIED	Infection-related
CUTI-520-49	*C. acnes* subsp. *acnes*	IA_1_	C1	PT	Hip	Infection undetermined/unlikely
CUTI-520-56	*C. acnes* subsp. *acnes*	IA_1_	C1	SF	Shoulder	Infection undetermined/unlikely
CUTI-519-25	*C. acnes* subsp. *acnes*	IA_1_	C9	SF	CNSD	Infection undetermined/unlikely
CUTI-539-27	*C. acnes* subsp. *acnes*	IA_1_	A1	PT	Spine	Infection-related
CUTI-210-3	*C. acnes* subsp. *acnes*	IA_1_	A1	SF	Hip	Infection-related
CUTI-208-3	*C. acnes* subsp. *acnes*	IA_1_	A1	SF	Spine	Infection-related
CUTI-255-21	*C. acnes* subsp. *acnes*	IA_1_	A1	PT	Spine	Infection-related
CUTI-255-26	*C. acnes* subsp. *acnes*	IA_1_	A1	SF	Spine	Infection-related
CUTI-238-35	*C. acnes* subsp. *acnes*	IA_1_	A1	SF	Hip	Infection undetermined/unlikely
CUTI-519-63	*C. acnes* subsp. *acnes*	IA_1_	A61	PT	Hip	Infection undetermined/unlikely
CUTI-241-40	*C. acnes* subsp. *acnes*	IA_1_	A1	SF	Knee	Infection-related
CUTI-236-73	*C. acnes* subsp. *acnes*	IA_1_	A1	SF	Spine	Infection undetermined/unlikely
CUTI-260-33	*C. acnes* subsp. *acnes*	IA_1_	A1	SF	Hip	Infection undetermined/unlikely
CUTI-260-32	*C. acnes* subsp. *acnes*	IA_1_	A1	SF	Knee	Infection-related
CUTI-251-63	*C. acnes* subsp. *acnes*	IA_1_	A1	SF	Hip	Infection undetermined/unlikely
CUTI-520-58	*C. acnes* subsp. *acnes*	IA_1_	A1	PT	CST	Infection undetermined/unlikely
CUTI-266-40	*C. acnes* subsp. *acnes*	IA_1_	A1	SF	Knee	Infection undetermined/unlikely
CUTI-544-67	*C. acnes* subsp. *acnes*	IA_1_	A1	SF	Shoulder	Infection undetermined/unlikely
CUTI-201-61	*C. acnes* subsp. *acnes*	IA_1_	A2	SF	Hip	Infection undetermined/unlikely
CUTI-521-13	*C. acnes* subsp. *acnes*	IA_1_	A1	SF	Ankle	Infection-related
CUTI-521-23	*C. acnes* subsp. *acnes*	IA_1_	A1	PT	Ankle	Infection-related
CUTI-538-56	*C. acnes* subsp. *acnes*	IA_1_	A2	SF	Knee	Infection undetermined/unlikely
CUTI-520-21	*C. acnes* subsp. *acnes*	IA_1_	A60	SF	Knee	Infection-related
CUTI-270-76	*C. acnes* subsp. *acnes*	IA_1_	A1	SF	Hip	Infection undetermined/unlikely
CUTI-250-33	*C. acnes* subsp. *acnes*	IA_1_	A1	SF	Spine	Infection-related
CUTI-250-69	*C. acnes* subsp. *acnes*	IA_1_	A1	SF	Spine	Infection-related
CUTI-520-59	*C. acnes* subsp. *acnes*	IA_1_	A1	SF	CNSD	Infection-related

* Phylotypes were determined according to Dreno et al. in the correlation table between MLST_8_ and SLST scheme [29]. ** ST—sequence type was determined according to SLST scheme. (SF—sonication fluid; PT—peri-implant tissue; CIED—cardiovascular implantable electronic device; VAD—ventricular assist device; CNSD—central nervous system device; CST—coronary stent).

## Data Availability

Generated raw reads were submitted to the Nucleotide Archive under the study accession number PRJEB67661.

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
