# Peer review of "Comparative Genomic Analysis of Cutibacterium spp. Isolates in Implant-Associated Infections"

_microorganisms, 2023, doi:10.3390/microorganisms11122971_

Round 1
Reviewer 1 Report
Comments and Suggestions for Authors
The findings of the study “Comparative Genomic Analysis of Cutibacterium spp. Isolates in Implant-Associated Infections” contribute to the existing body of knowledge on the sequence types and virulence factors of the bacterial species studied and could be of significance to the scientific audience in this area.
The manuscript is overall well-written, requiring a few minor corrections/edits and revisions. Please see my minor concerns below.
Minor Concerns:
Abstract, Lines 20-25: The 40 VFs were studied is which “selected” isolates? Later, in lines 214-215: the authors say the presence of 40 VFs was investigated in 64 Cutibacterium spp. Isolates. This contradicts with the information stated in the abstract. Were the 40 VFs meant to be “selected” for comparison instead? Please revise to remove the contradiction.
Lines 55-56: The timeline indicated by “…. reclassifications in last decades” is not clear (is it the last few decades or the last decade? Please provide the exact timeline.
Line 78: Use of “samples” in “…….prosthetic joint infections samples” is redundant. Remove “samples’ from the sentence.
Lines 87-89: The description of antibiotic susceptibility at the beginning of the paragraph seems out of place. From line 90, the narrative switches to description of virulence factors, and there is a lack of transition from the previous topic to the next. The description of antibiotic susceptibility/resistance will best fit in the first paragraph where the authors discuss why these bacteria are recognized as important opportunistic pathogens.
Lines 110-117: I assume the breakdown of isolates provided in this paragraph is for the same 64 isolates described in the first paragraph, lines 104-109, correct? However, the repetition is confusing without an explanation that the 64 isolates previously mentioned included the 51 isolates and 13 isolates obtained from two different sources. Please revise to clear potential confusion for the reader.
Results sections 3.1, 3.2, 3.3, 3.4: Scientific names of the bacterial species are not italicized on the narratives. Please italicize all scientific names in these sections.
Lines 218-220: The sentence starting with “Theis could potentially lead to misidenhomologs were identified……” does not make sense Please revise.
Comments on the Quality of English LanguageOverall well-written, with a few minor editing corrections and revisions needed. Please see my comments in the review report.
Reviewer 2 Report
Comments and Suggestions for Authors
The Authors proposed a work on comparative genomic analysis of 64 Cutibacterium spp. strains isolated from implant-associated infections, using whole genome sequencing.
The chosen topic is of scientific interest and is corresponding to the scope of the Journal Microorganisms.
The manuscript is well written and could be interesting and useful for other authors interested in studies such as molecular epidemiology of Cutibacterium spp strains.
The English style is fine but could be improved.
There are some minor inaccuracies to be corrected. I list some examples below.
Please, check and write in the cursive form all the microorganism names.
I want to point out that the Authors can mostly use “strains” instead of “isolates”.
Please, check the “et al.”. It should be written in cursive.
Line 12: 64 instead of “various”.
Line 151: counting instead of contigs.
Line 359: Please, correct “was found known”.
Line 365: Please, check and correct the phrase “…species-specific homolog was additionally…”
Line 376-378: Please, correct the style and add the strains of this work came from implant-associated infections.
Comments on the Quality of English LanguageThe English style is fine but could be improved in order to have a better quality of the manuscript.
